# Psychosocial Stress, Sedentary Behavior, and Physical Activity during Pregnancy among Canadian Women: Relationships in a Diverse Cohort and a Nationwide Sample

**DOI:** 10.3390/ijerph16245150

**Published:** 2019-12-17

**Authors:** Isabelle Sinclair, Myriane St-Pierre, Guillaume Elgbeili, Paquito Bernard, Cathy Vaillancourt, Sonia Gagnon, Kelsey Needham Dancause

**Affiliations:** 1Département des sciences de l’activité physique, Université du Québec à Montréal (UQAM), Montreal, QC H2X 1Y4, Canadabernard.paquito@uqam.ca (P.B.); 2Réseau intersectoriel de recherche en santé de l’Université du Québec (RISUQ), Laval, QC H7V 1B7, Canada; cathy.vaillancourt@iaf.inrs.ca; 3Douglas Hospital Research Center, Psychosocial Research Division, Montreal, QC H4H 1R3, Canada; guillaume.elgbeili@douglas.mcgill.ca; 4Research Center, University Institute of Mental Health at Montreal, Montreal, QC H1N 3V2, Canada; 5INRS Centre Armand-Frappier Santé Biotechnologie, Laval, QC H7V 1B7, Canada; 6Département d’obstétrique-gynécologie, Hôpital du Sacré-Coeur de Montréal, Université de Montréal, Montreal, QC H4J 1C5, Canada; gagnonsonia@hotmail.com

**Keywords:** distress, maternal and child health, mental health, physical activity, sedentarity, stress

## Abstract

Background: Past research shows that psychosocial stress and distress predict sedentary behavior and physical activity, but few studies focus on pregnant women. Our objective was to analyze relationships between psychosocial stress and distress with sedentary behavior and physical activity among pregnant women in Canada. Methods: We analyzed objectively-measured sedentary behavior and physical activity at 16–18, 24–26, and 32–24 weeks pregnancy in a sociodemographically diverse cohort of 70 women in Montreal, Canada. Participants completed the Perceived Stress Questionnaire and wore an accelerometer for 3 days that quantified sitting time and steps per day. We used univariate general linear models to analyze relationships between perceived stress with sedentary behavior and physical activity at each evaluation. To assess generalizability, we analyzed relationships between psychological distress with self-reported leisure-time sedentary behavior and daily energy expenditure in transportation and leisure physical activities among a sample representative of 166,095 women in the Canadian Community Health Survey. Results: In the Montreal cohort, we observed a positive association between perceived stress and sitting time, with small to moderate effect sizes (partial η^2^ = 0.08–0.16). We observed negative relationships between perceived stress and steps per day at the first two evaluations only, with small to moderate effect sizes (partial η^2^ = 0.08–0.11). Relationships for sedentary behavior were similar in the nationwide sample, but with smaller effect sizes (partial η^2^ = 0.02). There were no relationships between distress and physical activity in the nationwide sample. Conclusion: Psychosocial stress represents one risk factor for sedentarity, with relationships evident throughout pregnancy and at the population level. Relationships with physical activity are less consistent, but stress might represent a risk factor for low physical activity in early to mid-pregnancy. Results might guide the development of more comprehensive interventions targeting stress, sedentarity, and physical activity. In particular, integrating psychosocial health into interventions to reduce sedentarity, and including concrete guidelines on sedentary behavior in psychosocial health interventions, might be prioritized.

## 1. Introduction

The American College of Obstetricians and Gynecologists, the Society of Obstetricians and Gynaecologists of Canada, and other national committees emphasize the importance of maintaining regular physical activity (referring to body movement produced by skeletal muscles that results in energy expenditure [1]) and of initiating physical activity among women who had been inactive (that is, not meeting physical activity guidelines) prior to or early in pregnancy [2,3,4,5]. Some guidelines also include discussion of sedentary behaviors (defined as “any waking behavior characterized by an energy expenditure ≤1.5 metabolic equivalents while in a sitting, reclining or lying posture” [6]), noting the importance of avoiding a sedentary lifestyle during pregnancy. Except among women with severe complications, a daily program of 20–30 min of moderate-intensity exercise is suggested throughout pregnancy [4,5]. Despite these recommendations, past studies suggest that pregnant women spend at least 50% and up to 76% of waking hours in sedentary behavior [7,8,9], and studies from the US showed that women spend on average only 12 min per day in moderate physical activity [7]. In addition to cardiovascular and psychosocial health benefits, physical activity helps to promote safe weight gain during pregnancy, which reduces risk for gestational diabetes, preterm birth, and other adverse birth outcomes [2,3,4,5]. In concert, sedentary behavior and physical inactivity during pregnancy are associated not only with adverse cardiometabolic profiles among women, but with adverse infant outcomes such as macrosomia that might have long-term health implications [9]. Thus, promoting physical activity and limiting sedentary behavior should be prioritized throughout pregnancy.

Psychosocial stress might represent a risk factor for sedentary behavior and inactivity. Thus, the World Health Organization and other national and international organizations recognize stress as a key social determinant of health [10,11]. Stress refers to emotional and physiological reactions in response to a situation (a stressor) that requires the individual to react or adapt. When demands exceed available resources, the individual is likely to experience psychological distress [12,13]. Studies among the general population and among pregnant women show that psychosocial stress and distress are related to physical activity, although fewer studies of sedentary behaviors have been conducted. In the general population, greater symptoms of psychological distress, depression, and anxiety predict lower physical activity levels [14,15]. Similarly, our studies among the Canadian general population show that greater psychological distress predicted lower self-reported leisure-time physical activity and greater screen sedentary behavior [16]. Among pregnant women, past studies show that greater pregnancy-specific stress and perceived stress are associated with less exercise [17]. Others show that perceived stress is associated with physical inactivity during pregnancy [18]. Given the persistent high rates of physical inactivity and of sedentarity during pregnancy and the importance of promoting physical activity and limiting sedentary behavior for maternal and infant well-being, more studies examining correlates of these behaviors are necessary. Such studies might ultimately inform the development of more comprehensive intervention strategies, such as integrating psychosocial health into existing interventions to promote physical activity or to reduce sedentary behavior, and thus improve their efficacy.

We sought to assess relationships between psychosocial stress with sedentary behaviors and physical activity during pregnancy among Canadian women. We hypothesized that greater stress would be associated with more time spent in sedentary behaviors and with lower physical activity levels. We analyzed patterns among a cohort of pregnant women in Quebec using multiple measures of perceived stress and objectively measured sitting time and steps per day throughout pregnancy. We complemented these studies with analyses of patterns among a nationally representative sample of Canadian women to highlight population-wide trends. Together, these studies provide a portrait of the degree to which stress is related to sedentary behaviors and physical activity over the course of pregnancy, and how these relationships might generalize to the population. This could ultimately help to point to key areas for intervention to improve maternal and infant health.

## 2. Materials and Methods

This project was approved by the Research Ethics Committee of the Hôpital du Sacré-Coeur, Montreal (certificate number MP-32-2017-1373), by the University of Quebec in Montreal (certificate number 2015_e_788) and by Statistics Canada (project number 16-SSH-UQAM-4623). All participants provided written informed consent.

### 2.1. Montreal Cohort Study

#### 2.1.1. Sample

We recruited 81 pregnant women through the Department of Obstetrics and Gynecology at the Hôpital du Sacré-Coeur and associated clinics from February 2017 to December 2017 for detailed studies of stress and health behaviors during pregnancy. Recruitment was through informational flyers posted in waiting rooms and distributed by obstetricians. Eligible women were in their first trimester with singleton pregnancies. Exclusion criteria included multiple gestation, in vitro fertilization (IVF), plans to move away before delivery, cardiovascular conditions that could skew assessment of physiological measures of stress analyzed in our other studies, and inability to complete questionnaires in English or French. Hôpital du Sacré-Coeur was chosen as the primary recruitment site because of the diversity in clientele.

We collected data at three points during pregnancy: 16–18, 24–26, and 32–34 weeks’ gestation. Each assessment consisted of 3 days of data collection, typically three weekdays. Researchers met participants at a place of their choosing to drop off the questionnaires and equipment and returned after the third day to pick them up.

Of the 81 women, 5 suffered pregnancy loss and were excluded from the current analyses. A further 3 were lost to contact. Finally, data on sedentary behavior and physical activity were incomplete for 3 participants (Figure 1). The current sample includes 70 women with data on perceived stress and objectively-measured sedentary behavior and physical activity at each evaluation. No participants had medical contraindications for the practice of physical activity.

#### 2.1.2. Variables

Key variables included perceived stress, sedentary behavior (sitting time), and physical activity (steps per day). Perceived stress was assessed using the Perceived Stress Scale [19]. The questionnaire includes 14 questions on the degree to which life situations during the past month were appraised as stressful, with responses ranging from 0 (“Never”) to 4 (“Very often”). Response items are summed into a total score, with totals ranging from 0 (very low perceived stress) to 56 (very high perceived stress). Scores in the current sample ranged from 5 to 46. We used the Polar V800 watch (Lachine, QC, Polar Canada) to provide objective estimates of sedentary behavior and physical activity over the course of three days at each evaluation. Mean sitting time and steps per day over the course of the three days was computed for each evaluation and used in analyses.

Sociodemographic characteristics were assessed via questionnaire at each evaluation. Characteristics included participants’ age, education, income, country of origin, ethnicity, and maternal and pregnancy characteristics (number of children, due date). Household income was assessed using 10 categories ranging from less than $10,000 to more than $250,000 per year. This was reclassified into three categories (less than $20,000, $20,000–50,000, and more than $50,000) for descriptive statistics. Education was assessed using 7 categories, from “Secondary not completed” to “Post-doctorate”, with an open-ended option for other responses. This was reclassified into three categories (secondary or less, college, university or higher) for descriptive statistics.

#### 2.1.3. Analyses

We analyzed descriptive statistics, including means and standard deviations or frequencies, for each variable. We used repeated-measures general linear models to test changes in perceived stress, sedentary behavior, and physical activity over the course of pregnancy. We used univariate general linear models to analyze relationships between perceived stress (predictor) with sitting time and physical activity. Each model included age, number of children, education, income, immigration status (immigrant or non-immigrant), and ethnicity (visible minority or not a visible minority) as covariates. Analyses were conducted using SPSS version 22.0 (IBM Corp., Armonk, NY, USA).

### 2.2. Nationally Representative Sample

#### 2.2.1. Sample

The Canadian Community Health Survey (CCHS) [20] is an initiative of Statistics Canada to collect data on health status, health service utilization, and health determinants among the Canadian population. The sample includes 130,000 respondents aged 18 and over and 10,000 respondents aged 12–17 every two years. Data are collected using computer and telephone assisted interview software. Individual data are weighted to reflect the number of persons in the population represented by each respondent [20]. We analyzed data from the CCHS years 2011–2014. Our analyses included women aged 18 and older who reported being pregnant at the time of the questionnaire. The current sample includes 11,838 respondents representative of 166,095 women.

#### 2.2.2. Variables

Key variables included psychological distress, sedentary behavior (daily hours in leisure time sedentary behaviors), and physical activity (daily energy expenditure in transportation and leisure activities). The Kessler-10 Distress Scale [21] is a 10-item screening tool for nonspecific psychological distress. It encompasses symptoms of anxiety, depression, nervousness, and stress and is widely used in both clinical and nonclinical samples. Responses to each item are scored from 1–5 and summed, yielding a total score of 10–50 [22]. Scores of 30–50 indicate high risk of anxiety or depressive disorder, representing around 2% of the general population [22,23]. In the CCHS, scores were coded from 0 to 40 rather than 10 to 50. Because our goal was to analyze general psychosocial stress and distress, we excluded participants with scores indicative of high risk of severe mental disorder, as well as those who reported having anxiety or mood disorders, which represented 2.5% of the sample. Sedentary behavior was assessed through self-reported weekly hours of free time during the last three months on a computer, playing video games, watching television or videos, and reading [20]. Responses for all categories were summed and re-coded into hours per day. Physical activity was assessed by asking respondents to report the frequency and duration of participation in 21 leisure activities (with an additional option for other nonlisted activities), and biking or walking to work and school during the last three months. These responses were then used to calculate estimated energy expenditure in kilocalories per kilogram body weight per day. We chose this energy expenditure variable as our indicator of physical activity because it provided the most complete continuous physical activity variable representing both leisure and transport activity.

Sociodemographic characteristics assessed included participants’ age, number of children under age 18 living in the home, education, income, immigration status (immigrant or non-immigrant), and ethnicity. Household income was assessed via province-specific decile. For descriptive statistics, we reclassified these into the same 3 categories as in our Montreal cohort using Statistics Canada’s estimates of the upper income limit in each decile. The category “less than $20,000” included participants in Decile 1; the category “$20,000–50,000” included Deciles 2–6; and the category “more than $50,000” included Deciles 7–10. Education was assessed using 10 categories, from “Grade 8 or lower” to “University certificate beyond Bachelor’s degree”. This was reclassified into three categories (secondary or less, college, university or higher) for descriptive statistics.

#### 2.2.3. Analyses

We analyzed descriptive statistics, including means and standard deviations or frequencies, for each variable. Weighting each participant to represent population-wise data, we used univariate general linear models to analyze relationships between distress (predictor) with sedentary behavior and physical activity. Each model included age, number of children under age 18 living in the home, education, income, immigration status (immigrant or non-immigrant), and ethnicity (visible minority or not a visible minority) as covariates. Analyses were conducted using SPSS version 22.0 (IBM Corp., Armonk, NY, USA). 

In all analyses, *p*-values less than 0.05 were considered statistically significant.

## 3. Results

### 3.1. Descriptive Statistics

Sample size and means (SD) or frequencies for sociodemographic, perceived stress, and health behavior variables for the Montreal cohort are shown in Table 1, and those for the nationally representative sample are shown in Table 2. Analyses of changes over time in the Montreal cohort showed no linear trends in perceived stress (*p* = 0.132) or sedentary behavior (*p* = 0.408), but a trend toward decline in physical activity (*p* = 0.052) over the course of pregnancy (full analyses not shown).

### 3.2. Relationships Between Perceived Stress and Sedentary Behavior

In the Montreal cohort, perceived stress predicted greater sedentary behavior (sitting time) at all three evaluation points, (Table 3), with small to moderate effect sizes (partial η^2^ = 0.08–0.16). Perceived stress predicted physical activity at Evaluation 1 (partial η^2^ = 0.11) and 2 (partial η^2^ = 0.08) only. Figure 2 reflects estimated values for sitting time (a) and steps per day (b) at each evaluation period at low (5) and high (45) levels of perceived stress. 

A similar relationship with sedentary behavior was observed in the CCHS sample. Psychological distress predicted greater sedentary behavior (Table 4), but with much smaller effect sizes (partial η^2^ = 0.02). Psychological distress did not predict energy expenditure in leisure and transport physical activities in the CCHS sample.

## 4. Discussion

This study examined relationships between psychosocial stress and distress with sedentary behaviors and physical activity in two different samples. We chose the Perceived Stress Scale and the Kessler-10 Distress Scale because these questionnaires are widely used among both the general population and among pregnant women, which allows for comparison with other studies. These measures encompass many similar constructs but cannot be directly compared among the two samples. Similarly, we cannot directly compare objectively measured sitting time and steps per day in the Montreal cohort to self-reported time spent in sedentary leisure activities and estimated energy expenditure in leisure and transport activities in the CCHS. However, comparison of the general patterns provides a broad overview of relationships between psychosocial stress with sedentarity and physical activity that helps to contextualize results and that points to areas for future research.

### 4.1. Sedentary Behavior

Sitting time in the Montreal cohort averaged more than 8 h per day. Similar accelerometer-based studies in the US showed estimates of 424 min (around 7 h) per day in sedentary behavior, with only minor variations by trimester [7]. Others showed estimates of 12.4 h per day at gestation week 18 and 12.9 h per day at week 35 [8]. Studies in Spain were similar, with women spending 10.0 h per day in sedentary behavior, and no notable differences between weekday (10.1 h/day) and weekend (9.9 h/day) estimates [24]. Calculation of the percent of time spent in sedentary behavior depends on the bout length of interest, which complicates the direct comparison of accelerometer-measured sedentary behaviors across studies [25]. Nevertheless, results of the studies cited above are consistent in showing that pregnant women spend a large percentage of their waking hours in sedentary behavior.

Self-reported participation in leisure-time sedentary behaviors in the CCHS was also consistent with other studies, averaging 3.3 h per day. This is similar to our estimates of screen sedentary behavior among men and women in the general population, which ranged from around 18 to 25 h per week (2.6–3.6 h/day) and varied by gender and sociodemographic characteristics [16]. Among pregnant women, studies in the US that assessed responses to 2 questions on time spent watching TV and sitting quietly performing other activities showed an average of 2.6 h of sedentary behavior per day [26]. Studies in Spain using the Sedentary Behavior Questionnaire (SBQ) showed that women self-reported around 8.7 h of sedentary behavior per day [24]. The SBQ assesses 11 sedentary behaviors including leisure (e.g., television, playing a musical instrument), transport (e.g., driving or traveling in a motor vehicle), rest, and work activities, so estimates are higher than for leisure activities alone.

In both samples, our results show relationships between psychosocial stress and sedentary behavior, and these relationships persisted when controlling for important sociodemographic covariates. In the Montreal cohort, greater perceived stress was associated with greater sitting time at 16–18, 24–26, and 32–34 weeks pregnancy, with small to moderate effect sizes. Similarly, greater psychological distress predicted greater leisure-time sedentary behavior among pregnant women in the general Canadian population. Effect sizes were much smaller in the CCHS, highlighting the great deal of variability at the nationwide level. Furthermore, imprecisions in self-reported estimates of leisure-time physical activity might weaken the relationships observed. Nevertheless, results highlight overall that the level of sedentarity among pregnant women in Canada is high, and that stress and distress represent risk factors for these patterns.

These relationships are consistent with results among adults in other countries. For example, among Belgian adults, symptoms of psychological distress predicted greater sitting time [27], and among US Latino adults, chronic and lifetime stressors predicted sedentary behavior [28]. Similar results have been observed for screen sedentary behavior. Studies among Australian women in socioeconomically disadvantaged neighborhoods showed that psychological distress predicted increased television viewing [29], and studies among Puerto Rican adults living in the US showed that those with the highest perceived stress had greater television viewing time [15]. Our own studies among Canadian adults showed positive relationships between psychological distress and sedentary behavior among all adults, with variations by sociodemographic characteristics such as ethnicity, immigration status, and education [16]. However, we identified very few similar studies among pregnant women. Researchers in Brazil observed no associations between perceived stress during pregnancy and low levels of physical activity derived from self-report responses on the International Physical Activity Questionnaire. In fact, lower levels of anxiety were associated with higher rates of physical inactivity. The authors note that women’s physical activity in the setting revolves largely around childcare and housework, which might be a source of anxiety [30]. Results highlight the need for more studies of relationships between mental health and sedentary behavior among pregnant women in different settings.

### 4.2. Physical Activity

In the Montreal cohort, steps per day averaged 7878 in early pregnancy and 6273 in late pregnancy. These figures are consistent with accelerometer data among a sociodemographically diverse sample of women at less than 25 weeks of pregnancy in Norway, which showed step counts ranging from 7718 to 9603 depending on ethnicity [31]. Similar studies among Caucasian women in Spain assessed at 16 weeks’ gestation showed an average of 7745 steps per day [32]. Systematic reviews highlight that both the frequency and intensity of physical activities tend to decline over the course of pregnancy [33], consistent with the trend toward decline from Evaluation 1 to Evaluation 3 observed in our study.

Energy expenditure in the CCHS sample averaged 1.6 kcal/kg/day. This is consistent with observations among pregnant women in England, which showed mean leisure energy expenditure, based on estimations from self-report questionnaires, of 1.9 kcal/kg/day in the first trimester, 1.3 in the second trimester, and 1.2 in the third trimester [34]. These values are consistent with moderate leisure activity levels (defined as daily energy expenditure of 1.5 to <3.0 metabolic equivalents) [35].

Relationships between stress or distress and the measures of physical activity analyzed were less consistent than relationships with sedentary behavior. In the Montreal cohort, relationships between perceived stress and steps per day were significant in early and mid but not in late pregnancy. The lack of a significant relationship in late pregnancy would be expected to weaken any associations in analyses including women in all stages of pregnancy, and indeed, there were no relationships between distress and energy expenditure in the CCHS sample. Other population-wide studies of psychosocial health and physical activity have shown mixed relationships. For example, studies from the US showed that perceived stress was associated with less leisure time physical activity [15,36]. Similarly, psychological distress predicted decreased likelihood of participating in medium or high levels of leisure physical activity among Australian women in socially disadvantaged neighborhoods [29]. Our studies among Canadian adults showed very weak negative relationships between psychological distress and physical activity, with more marked negative relationships among adults with greater education and income [16]. In general, results of the current study suggest that stress and distress might represent risk factors for less physical activity in early or mid-pregnancy, but not in late pregnancy. Other risk factors likely play a more important role in the decline in physical activity over the course of pregnancy and the low levels of physical activity observed in late pregnancy in many studies.

### 4.3. Strengths, Limitations, and Future Directions

Study strengths in the case of the Montreal cohort include the use of a validated questionnaire for assessing perceived stress, and objectively-measured sedentary behavior and physical activity. Furthermore, the assessment of perceived stress three times during pregnancy allowed us to highlight relationships over the course of pregnancy. Finally, the study is strengthened by the diverse sample, including immigrant women and women with low income who often face barriers in participating in research studies [37]. Major limitations of the cohort study include the sample size, which is too small to test advanced statistical models. The current analyses were not corrected for multiple testing, and studies in larger cohorts are needed to replicate the results. Furthermore, the indicators of physical activity and sedentary behavior, chosen to allow comparison to other studies, do not take into account many characteristics of interest. For example, our measure of physical activity does not reflect the intensity of activity. Similarly, our measure of sedentary behavior includes total sitting time throughout the day, but we cannot account for the type of activity. Important differences have been noted in relationships between mental health and sedentary activities that are mentally active (such as office work) versus passive (such as watching television) [38]. Finally, sedentary behaviors and physical activity might vary on weekend versus weekdays, but measures here represent primarily weekdays only.

Strengths of the CCHS analyses include the use of a validated questionnaire to assess psychological distress, and the large representative sample that permits generalization to the population level. The study is limited by the self-reported data collection, which likely underestimates sedentary behavior and overestimates physical activity compared to objective measures. Furthermore, our indicator of physical activity is based on estimated values of energy expenditure for common activities that, while widely used, provide only a basic estimate of actual leisure and transportation energy expenditure. For both sedentary behavior and physical activity, the lack of data on activities outside of leisure and transportation is a major limitation.

In both studies, our cross-sectional analyses do not allow us to identify causal relationships. Furthermore, we cannot control for a number of confounding variables associated with sedentary behavior and physical activity. Data on season of data collection, physical characteristics such as body weight and weight gain during pregnancy, medical conditions that might be associated with restricted activity levels, health behaviors such as dietary patterns and smoking, and pre-pregnancy characteristics are not available or comparable between datasets. We thus focused on sociodemographic covariates that could be compared between datasets and that have been associated with stress, sedentary behaviors, or physical activity in past studies. Physical, medical, and health behavior characteristics might mediate or moderate relationships between stress and sedentary behavior or physical activity and should be integrated into future studies. Such analyses might point to key characteristics that could be targeted to reduce the relationship between distress and sedentary behavior.

In addition to assessment of confounding variables, future studies would also benefit from testing relationships between stress and sedentary behaviors or physical activity in larger and more diverse samples. Patterns might be more pronounced, for example, among women with very high stress levels, among socially disadvantaged women, or in communities where opportunities to participate in leisure and transportation physical activity are limited. Such analyses might point to groups among whom relationships are particularly pronounced and thereby guide future interventions.

## 5. Conclusions

Stress, physical activity, and sedentary behaviors are all highlighted as major determinants of maternal and child health [39]. Despite existing recommendations and interventions, the level of sedentarity during pregnancy remains high, and physical activity levels remain low. Our results suggest that stress is a predictor of sedentary behavior throughout pregnancy. Relationships with physical activity, on the other hand, are not evident in late pregnancy. Current guidelines focus strongly on physical activity during pregnancy, with little direct advice for pregnant women on sedentary behaviors. Sensitizing pregnant women to the importance of sedentary behavior and integrating this information into existing programs to reduce stress during pregnancy might be promising. Similarly, integrating psychosocial health into interventions to reduce sedentary behavior throughout pregnancy and interventions to promote physical activity in early-to mid-pregnancy might be prioritized. Pregnancy represents a key period for promoting both healthy behaviors and psychosocial health, and such interventions could have long-term impacts on health and wellbeing for both women and their children.

## Figures and Tables

**Figure 1 ijerph-16-05150-f001:**
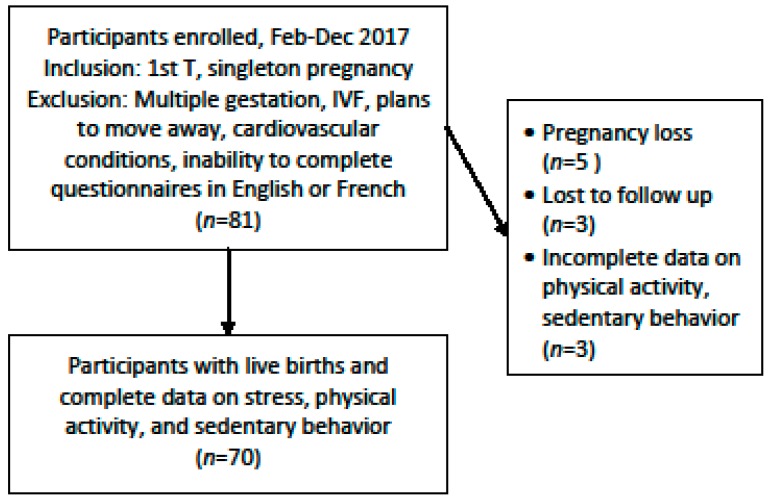
Sample characteristics, Montreal cohort.

**Figure 2 ijerph-16-05150-f002:**
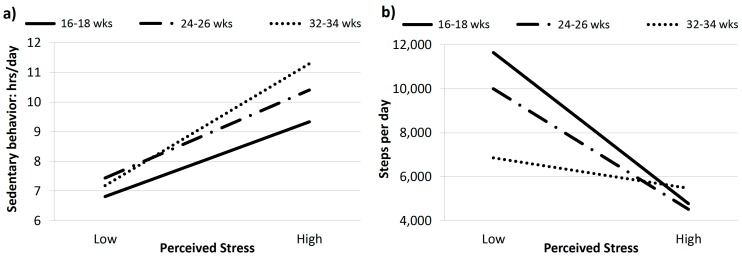
Relationships between perceived stress with (**a**) sedentary behavior (hours per day) and (**b**) physical activity (steps per day) at each evaluation period (16–18, 24–26, 32–34 weeks pregnancy) in the Montreal cohort.

**Table 1 ijerph-16-05150-t001:** Descriptive statistics: Means (SD) or frequencies for key variables in the Montreal cohort.

Variable	Mean/Frequency	Range
Age	31.1 (5.9)	19–45
Number of children	0.9 (1.1)	0–5
Household Income, *n* (%)		
<$20,000	20 (28.6)	
$20,000–50,000	35 (50.0)	
>$50,000	15 (21.4)	
Education, *n* (%)		
Secondary	21 (30.0)	
College	16 (32.9)	
University	33 (47.1)	
Immigrant, *n* (%)	46 (65.7)	
Visible minority, *n* (%)	48 (68.6)	
Perceived Stress		
Evaluation #1	24.4 (7.0)	7–44
Evaluation #2	22.2 (6.7)	6–37
Evaluation #3	22.0 (7.7)	5–42
Sedentarity (sitting time, h/day)		
Evaluation #1	8.0 (1.9)	3.8–13.7
Evaluation #2	8.7 (1.8)	5.7–13.4
Evaluation #3	8.9 (2.1)	4.3–14.5
Physical activity (steps/day)		
Evaluation #1	7878 (4381)	1464–26,672
Evaluation #2	7644 (3566)	1948–19,842
Evaluation #3	6273 (3136)	593–15,762

**Table 2 ijerph-16-05150-t002:** Results of general linear models testing relationships between perceived stress with sedentary behavior and physical activity, Montreal cohort. Significant values in bold.

Variable	Sedentary Behavior			Physical Activity		
	β	*p*-Value	η^2^	β	*p*-Value	η^2^
Evaluation #1 (16–18 weeks pregnancy)						
Age (years)	−0.04	0.358	0.01	62.65	0.580	<0.01
No. children	0.24	0.344	0.01	317.86	0.597	<0.01
Education	**0.23**	**0.045**	**0.06**	−446.81	0.101	0.04
Household income	−0.39	0.256	0.02	232.22	0.779	<0.01
Immigration status	−1.00	0.126	0.04	417.52	0.790	<0.01
Visible minority status	0.50	0.426	0.01	−386.47	0.797	<0.01
Perceived stress	**0.10**	**0.005**	**0.12**	**−227.98**	**0.007**	**0.11**
Evaluation #2 (24–26 weeks pregnancy)						
Age (years)	0.00	0.955	<0.01	43.55	0.623	<0.01
No. children	0.19	0.420	0.01	823.12	0.087	0.05
Education	0.05	0.606	<0.01	−350.04	0.089	0.05
Household income	−0.40	0.227	0.02	132.00	0.840	<0.01
Immigration status	0.03	0.964	<0.01	−268.16	0.823	<0.01
Visible minority status	0.74	0.219	0.02	958.38	0.419	0.01
Perceived stress	**0.07**	**0.024**	**0.08**	**−** **139.15**	**0.028**	**0.08**
Evaluation #3 (32–34 weeks pregnancy)						
Age (years)	0.00	0.972	<0.01	6.87	0.933	<0.01
No. children	0.02	0.936	<0.01	439.23	0.318	0.02
Education	0.00	0.975	<0.01	231.63	0.222	0.02
Household income	−0.48	0.218	0.02	709.02	0.250	0.02
Immigration status	−0.30	0.676	<0.01	−278.83	0.807	<0.01
Visible minority status	0.31	0.661	<0.01	−1226.89	0.268	0.02
Perceived stress	**0.11**	**0.001**	**0.16**	−49.47	0.341	0.01

**Table 3 ijerph-16-05150-t003:** Descriptive statistics: Means (SD) or frequencies for key variables in the Canadian Community Health Survey sample.

Variable	Mean/Frequency
Age (Years)	30.2 (5.7)
Number of people <18 years old living in the household	0.8 (0.9)
Household Income, %	
<$20,000	11.1
$20,000–50,000	42.5
>$50,000	46.4
Education, %	
Secondary	15.7
College	42.3
University	42.0
Immigrant, %	35.0
Visible minority, %	34.3
Psychological Distress	4.0 (3.4)
Sedentarity (leisure activities, h/day)	3.3 (2.0)
Physical activity (daily energy expenditure, transportation & leisure activities, kcal/kg/day)	1.6 (1.6)

**Table 4 ijerph-16-05150-t004:** Results of general linear models testing relationships between psychological distress with sedentary behavior, Canadian Community Health Survey sample. Significant values in bold.

Variable	Sedentary Behavior			Physical Activity		
	β	*p*-Value	η^2^	β	*p*-Value	η^2^
Age (years)	0.00	0.905	0.00	0.00	0.908	<0.00
No. children in household	−0.30	0.058	0.02	0.03	0.730	<0.00
Education	**−0.18**	**0.002**	**0.03**	0.08	0.055	0.01
Household income	0.03	0.306	<0.01	0.00	0.994	<0.00
Immigration status	−0.25	0.252	<0.01	**0.37**	**0.045**	**0.01**
Visible minority status	0.01	0.582	<0.01	0.00	0.899	<0.00
Perceived stress	**0.09**	**0.001**	**0.02**	0.01	0.580	<0.00

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
