# Peer review of "Psychosocial Stress, Sedentary Behavior, and Physical Activity during Pregnancy among Canadian Women: Relationships in a Diverse Cohort and a Nationwide Sample"

_ijerph, 2019, doi:10.3390/ijerph16245150_

Round 1
Reviewer 1 Report
Overall Comments:
The authors present a diverse cohort and a nationwide sample studies “to analyse relationships between psychosocial stress and distress with sedentary behavior among pregnant women in Canada” (Page 1, lines 17-19).
They report, “relationships between psychosocial stress and sedentary behavior” (page 7, lines 205-6). In addition, they affirm “level of sedentarity among pregnant women in Canada is high, and that 213 stress and distress represent risk factors for these patterns” (lines 213-14), but the main conclusions do not include these statements and refer to more global aspects.
I found the study sufficiently designed, but some methodological clarifications are necessary:
The ethical committee of the Sacré-Coeur hospital (p 2, 69-71) approved the study, but it is not indicated if the study was registered in the ethical commission of the participating universities (UQAM, INRS and Université de Montréal). The reason for the reduced sample is not clarified. No information is provided on how many pregnant women were proposed, how many accepted, how many were rejected because the inclusion criteria The information provided by the heart rate monitor is clearly insufficient to determine sedentary behavior (p. 3, 91-93), being interesting the use of complementary measures such as the number of steps or the written log of the activity carried out on the days analyzed. The characteristics selected to evaluate the linear models (sociodemographic characteristics) do not include conditioning elements for the performance of physical activity such as overweight, healthy habits (smoking, drugs, eating) or medical contraindications for the practice of physical activity, which could even be considered as inclusion criteria. More information is needed on the number of pregnant women who answered the CCHS questionnaire.
The document needs some revisions to be accepted. My recommendation is “Reconsider after major revision”.
Specific Comment:
It is recommended to clarify the conclusions and point them towards the objective.
Is the study registered in the ethical commission of the respective universities?
Please provide a flowchart of the cohort study participants and the number of respondents who were pregnant women.
Please clarify how information on daily activity and the sedentary behavior index is obtained.
Consider physical, health and personal characteristics to evaluate the linear models.
Reviewer 2 Report
General Comments:
The present study aimed to analyze relationships between psychosocial stress and distress with sedentary behavior among pregnant women in Canada. The authors have presented a well-written manuscript with coherence between sections. The present study is original, the research purpose is relevant, the rationale is very clear, and the experimental protocol is well-designed. The findings are interesting and provide novel and useful information about perceived stress and sedentary behaviour among pregnant women. There are some minor points that need to be addressed before the manuscript may be considered for publication in order to improve the comprehension of the information provided as listed in “Specific Comments” below.
Specific Comments:
Introduction:
Please add a clear definition of stress and sedentary behaviour and support it with references. Page 1, line 39: Please also add the original ACOG reference (doi: 10.1097/AOG.0000000000001214). Page 2, line 67: Please add the hypothesis for the present study.
Material and methods:
Page 2, line 84: Please provide description of the sample size calculation or a rationale on how the authors decided for 70 women. Page 3, line 95: Was pre-pregnancy physical activity level taken into consideration? Page 3, line 109: Please provide the accepted α-value for the analysis. Page 3, line 109 and 248: Please describe which post-hoc test was chosen for the analysis.Discussion:
Page 7, line 193: please provide references to support the statement.Page 8, line 250: I would like to suggest to the investigators to add the directions for future studies.
Round 2
Reviewer 1 Report
I am pleased to have been able to contribute to the improvement of the article submitted by the author.
It can be seen in the manuscript sent that the comments made by this reviewer have been accepted and adapted, so it only remains to congratulate the author on the document presented and trust that it is a valuable contribution in the field